# Reproducibility Study of "Cooperation, Competition, and Maliciousness: LLM-Stakeholders Interactive Negotiation"

**Jose L. García**\*, **Karolína Hájková**\*, **Maria Marchenko**\*, **Carlos Miguel Patiño**\*
*University of Amsterdam*
*{jose.garcia.carrillo, karolina.hajkova2, maria.marchenko, carlos.patino.paz}@student.uva.nl*

**Reviewed on OpenReview:** *https://openreview.net/forum?id=MTrhFmkC45*

## Abstract

This paper presents a reproducibility study and extension of "Cooperation, Competition, and Maliciousness: LLM-Stakeholders Interactive Negotiation." We validate the original findings using a range of open-weight models (1.5B-70B parameters), GPT-4, and GPT-4o Mini while introducing several novel contributions. We analyze the Pareto front of the games, propose a communication-free baseline to test whether successful negotiations are possible without agent interaction, evaluate recent small language models' performance, analyze structural information leakage in model responses, and implement an inequality metric to assess negotiation fairness. Our results demonstrate that smaller models (<10B parameters) struggle with format adherence and coherent responses, but larger open-weight models can approach proprietary model performance. Additionally, in many scenarios, single-agent approaches can achieve comparable results to multi-agent negotiations, challenging assumptions about the necessity of agent communication to perform well on the benchmark. This work also provides insights into accessibility, fairness, environmental impact, and privacy considerations of LLM-based negotiation systems.

## 1 Introduction

The rapid advancement of Large Language Models (LLMs) has opened new opportunities for Artificial Intelligence (AI) applications, particularly in the form of autonomous AI agents. These agents, capable of interacting and communicating with one another, can be especially useful in negotiation systems. Negotiations involve complex multi-agent interactions that require resolving multi-issue scenarios with little to no supervision, so it is critical to evaluate whether these agents are reliably performing their intended tasks (Shavit et al., 2023). Within this context, reproducibility studies are valuable, as they help validate benchmarks the AI community can use to evaluate the performance of current and future models (Reuel et al., 2024; Chang et al., 2023). This paper aims to reproduce the results of an LLM testbed specially designed for negotiation games (Abdelnabi et al., 2024) to contribute to developing robust benchmarks.

Our reproducibility goal is to assess whether the benchmark effectively evaluates the models' negotiation performance and whether the Chain-of-Thought (CoT) prompt configurations contribute to successful deal-making, especially in terms of accountability (Chan et al., 2024) and confidentiality. To achieve this, we compared different open-weight models against a state-of-the-art (SOTA) closed-source model. Building on the original paper, we analyze the Pareto front of the proposed games and establish a strong baseline to test whether an agent can successfully propose a deal without communicating with other parties.

We also present extensions that address the environmental impact, accessibility, fairness, and confidentiality. For environmental impact and accessibility, we use the benchmark to evaluate open-weight models and compare their performance to larger, more energy-consuming ones. Smaller models reduce the computational resource requirements (Fu et al., 2024; Sinha et al., 2024), and we compare the performance between models

---

*Equal contribution

that range from 1.5B to 70B parameters. For fairness, we introduce an inequality metric to assess whether all parties benefit equally from the negotiation. For confidentiality, we compare open-weight models with a closed-source model that requires access through an external API and may expose attack surfaces that can be exploited by malicious actors (Evertz et al., 2024; Dunn et al., 2024).

## 2 Scope of Reproducibility

We focus our reproduction efforts on investigating the following claims from the paper:

1. Open-weight models fall slightly behind SOTA closed-source models.
2. Small models—i.e., models with 8B parameters or less—are not suitable for the negotiation benchmark because they cannot follow the format of the game.
3. Agents have better outcomes when using CoT structures in their prompts.
4. The agents' incentives in the benchmark result in non-zero-sum games.
5. The benchmark measures cooperation, communication, and negotiation skills.

The code and instructions for reproducing the results of this work, along with our additions and the fixes mentioned in Section A.9, are available on Github [1].

## 3 Original Paper

### 3.1 Benchmark and Base Game

The original paper (Abdelnabi et al., 2024) introduces a benchmark to evaluate whether LLMs can successfully reach agreements in a multi-agent, multi-issue, semantically rich negotiation game over multiple rounds. The benchmark is designed to test agents in a dynamic, multi-turn scenario that requires advanced reasoning and strategic decision-making. Specifically, it assesses their ability to reach agreements through arithmetic calculations, inference, exploration, and planning, in addition to essential skills such as communication, collaboration, and instruction following.

The benchmark's base negotiation game is adapted from HarborCo (Susskind, 1985), a game traditionally used to teach negotiation skills involving six parties negotiating over five key issues related to constructing a sports complex. Each party is assigned a name and a role with different goals and preferences for each issue. That way, every issue is presented with three to five options, each assigned with a score for the evaluating agent. A deal proposal is agreeable for an agent if the sum of the chosen options across all issues meets or exceeds its minimum utility score threshold.

Before the game begins, all agents are given a global context outlining the negotiation setup, including information about the issues and the other agents' assigned names, roles, objectives, and preferences. The game then encourages agents to cooperate and compromise when necessary over the negotiation rounds to achieve a successful outcome, as a negotiation is only considered successful if at least five out of the six players surpass their scores threshold and accept the final deal, including Player 1 ($p_1$) and Player 2 ($p_2$).

Player 1 and Player 2 are specially designated roles with veto power: for the final deal to be accepted, it must receive approval from both. $p_1$, referred to as *the main negotiator*, is also responsible for proposing the negotiation's first and final deals. As the main negotiator, $p_1$ receives a bonus of 10 points upon suggesting a final deal accepted by everyone.

The initial deal is provided to $p_1$ at the start of the game, serving as the foundation for the follow-up discussions, and the remaining five agents aim to negotiate better deals for themselves. If the parties fail to reach a valid agreement, each agent receives their Best Alternative To a Negotiated Agreement (BATNA) score. BATNA typically corresponds to the minimum utility score threshold of each party but may vary depending on the game variation.

---

[1]https://github.com/cmpatino/llm_negotiation

### 3.2 Game Variations

To further expand the benchmark, the authors introduced variations of the base game where a single agent exhibits greedy or adversarial behaviors, influencing interactions and outcomes. In these variations, an agent can be designated as greedy, focusing solely on maximizing its own benefit, or adversarial, actively targeting another agent to sabotage the deal.

The adversarial behavior is also divided into two variations: targeted, where the adversarial agent is explicitly assigned a specific target, and untargeted, where the agent independently selects which opponent to sabotage.

LLMs are evaluated even further across four difficulty levels: *Base, Game 1, Game 2, and Game 3.* These variations adjust the base game's difficulty, with Games 1 and 3 being more challenging and Game 2 presenting the highest level of difficulty. Each variation introduces a new and unique setup with modified negotiation roles, objectives, and minimum utility thresholds designed to test the robustness of LLMs across diverse scenarios.

### 3.3 Evaluation Metrics

To systematically evaluate agent performance on all of these game variations, the benchmark introduces metrics to quantify how well agents align with their assigned roles and objectives, as well as their adherence to the negotiation's rules. These metrics include:

- *5/6-way* and *6-way*: The percentage of experiments in which the final deal is accepted by at least five or all six parties.
- *Any*: The percentage of experiments in which there exists a deal accepted by at least five parties at any point in the negotiation.
- *Wrong*: The percentage of rounds in which an agent proposed a deal below its own score threshold.
- *Leakage of Information*: The percentage of rounds in which an agent revealed their reasoning or private information to other parties.

We used the 5/6 way and 6-way accepted deals metrics as our main method of comparing the models, using their values after the final round of negotiation. As mentioned before, the only agent able to propose final deals is $p_1$. Other models are not prompted to vote, as voting happens automatically: if the suggested deal meets the minimum score of the agent it automatically votes for the deal. This setup creates a significant power imbalance between the models: as only $p_1$ can suggest deals, and the others accept or decline them automatically, their only way of influencing the outcome of the game is trying to convince $p_1$ of suggesting a more preferable deal for them.

## 4 Methodology

### 4.1 Models

The original paper compares the benchmark on different open- and closed-weight LLMs. To reproduce the experiments and validate their claims, we selected a range of models that performed well on common evaluation metrics (Chiang et al., 2024). Given the concerns about reproducibility with proprietary models (Palmer et al., 2024), our budget and computational resource constraints, we opted for open-weight models, as detailed in Table 1.

The original paper focused on using GPT-4 (gpt-4-0613) (OpenAI, 2023), so we show the results for that model on the base game. We also included GPT-4o Mini (gpt-4o-mini-2024-07-18)(OpenAI, 2024) in our analysis to include a model from the same provider as GPT-4 that we expected to perform well on the benchmark based on public metrics (Chiang et al., 2024). Additionally, GPT-4o Mini is around 100 times cheaper to run than GPT-4 and has similar or better performance on this benchmark than GPT-4 as shown by Tables 2 and 5. Because of the performance comparison to GPT-4 and significantly lower cost, we chose to run the majority of our analysis with GPT-4o Mini.

| Model | Open Weights | Size | Release Date | Context size | GPQA | MMLU | Energy, J |
|---|---|---|---|---|---|---|---|
| GPT-4 | ✗ | ~1.8T* | Jun 2023 | 32k | 35.7% | 86.4% | – |
| GPT-4o Mini | ✗ | ~8B* | Jul 2024 | 128k | 40.2% | 82.0% | – |
| Llama 3.3 70B | ✓ | 70B | Dec 2024 | 128k | 50.5% | 86.0% | 800k |
| Llama 3.0 70B | ✓ | 70B | Apr 2024 | 8k | 39.5% | 82.0% | 600k |
| Llama 2 13B | ✓ | 13B | Jul 2023 | 4k | 22.3% | 47.8% | 2000k |
| Llama 3.1 8B | ✓ | 8B | Jul 2024 | 128k | 30.4% | 69.4% | 1000k |
| Qwen 2.5 7B | ✓ | 7B | Sept 2024 | 128k | 36.4% | 74.2% | 100k |
| Qwen 2.5 1.5B | ✓ | 1.5B | Sept 2024 | 128k | 24.2% | 60.9% | 20k |
| Qwen2.5 32B Instruct | ✓ | 32B | Sept 2024 | 131k | 49.5% | 83.3% | 150k |
| Phi-4 | ✓ | 14B | Dec 2024 | 16k | 56.1% | 84.8% | 600k |
| DeepSeek-R1-Distill 7B | ✓ | 7B | Jan 2025 | 128k | 49.1% | – | 150k |

Table 1: Models used for reproducing the original study of the paper. The number of parameters from GPT-4 and GPT-4o Mini is an estimation based on Abacha et al. (2024). GPQA is a Graduate-Level Google-Proof Q&A Benchmark. MMLU is the Massive Multitask Language Understanding benchmark. The benchmarks for the latest DeepSeek were not reported. We report the consumed energy for one game, and we provide an analysis of energy consumption in Section A.4.

## 4.2 Reproduction Setup

We used the code provided by the authors[2] to reproduce the results in the original paper. While the authors' repository included much of the necessary code, the specific prompts used in the ablation study were not accessible. Therefore, we inferred the relationships between the available prompts and their corresponding ablation configurations. Restructuring the cooperative scratchpad prompts (Section A.10) and parametrizing the program's interface to select between different CoT prompts, we enabled an easier reproduction of the ablation study in Table 3 of the original paper.

## 4.3 Extensions

### 4.3.1 Pareto Analysis

We use Pareto front membership as a binary indicator of a deal's quality.

**Definition 1:** Given two deals $d_1$ and $d_2$, we say that $d_1$ *Pareto dominates* $d_2$ if, for at least one party, the score in $d_1$ is strictly better than in $d_2$, and for all parties, the score in $d_1$ is not worse than in $d_2$.

**Definition 2:** The *Pareto front* is the set of deals not Pareto dominated by any other deal.

### 4.3.2 Single-agent Baseline

The original paper proposes a multi-agent setup, as described in Section 3.1, where agents communicate and negotiate over multiple rounds to refine the initial deal. However, we observed that the global instructions given to the agents before the game begins contain crucial information regarding the overall setup, including information related to other agents.

We argue that, based on the names of the other agents, such as *Department of Tourism* and *Environmental League*, along with the provided descriptions for each issue and their options, the main negotiator $p_1$ can infer enough information to develop an understanding of the other agents' motivations and goals without direct communication. We provide a detailed example of the information available to $p_1$ from the game's description in Section A.1.

We developed two scenarios to evaluate $p_1$'s ability to generate deals independently. The first, *Single-agent 1 call*, is a simplified version where $p_1$ suggests the given initial deal to himself, then reasons for a single

---

[2]https://github.com/S-Abdelnabi/LLM-Deliberation

round, and proposes the final deal immediately. The second, *Single-agent 6 calls*, is an extended version that allows the agent to prompt itself iteratively over six rounds. In this setup, the agent continuously refines and expands its previous reasoning and planning before proposing a new deal.

We tested this new baseline across most of the game variations, including greedy agents and Games 1, 2 and 3. We expected this baseline to perform well in the compromising variant but poorly in the greedy variant, as the latter requires an agent to act beyond its initial prompt specifications. We did not include the adversarial game variant since it relies on agents communicating to achieve their adversarial role.

In the original paper, the authors already explored a configuration without direct communication between agents. However, this setup differs from our proposed baseline. In their approach, the initial deal is generated randomly, and then each agent, without communicating, improves the deal independently based on its own hidden score before passing it to the next agent. Despite not communicating, there is still some form of interaction in this setup, as each agent improves the deal for itself before passing it. In contrast, our setup demonstrates how much information a single agent can infer from the open game rules.

We evaluate this baseline using the two best-performing models on the multi-agent original setup: the proprietary GPT-4o Mini and open-weight Qwen 2.5 32B (Yang et al., 2024).

### 4.3.3 Structure Leakage

Open LLMs face more challenges to adhere strictly to instructions than proprietary models (Gudibande et al., 2023), which poses a problem in games requiring strict instruction obedience for formatting rules. In the negotiation game, LLMs are prompted to generate a response (`full answer`) with specific tags in place to distinguish between their `plan`, `scratchpad`, and `public answer`. If the agent fails to separate its output correctly, parts of its private output, like the CoT about other agents or future plans, are shared with all agents. This information leakage is possible due to the modeling decisions in the original implementation.

To measure this structural leakage of information, we check whether 1) the `full answer` and `public answer` differ, 2) at least one of the `<PLAN>`, `</PLAN>`, `<SCRATCHPAD>`, `</SCRATCHPAD>` tags is present in the public answer, or 3) tags `<DEAL>` or `</DEAL>` are missing on the public answer. If either of these conditions is met, the agent did not conform to the instructions, implying the unwanted release of information. We report this metric because sharing the `full answer` of the agents disrupts the negotiation dynamics and introduces unrealistic scenarios that could be avoided using different implementation decisions.

On the original paper, the leakage of information metric was measured using a LLM agent, GPT-4, to check the answers of the agents after each round and check the percentage of answers with leaked private information. We compare our structure-induced leakage scores to the scores of the original leakage metric, using GPT-4o Mini instead.

### 4.3.4 Inequality Metric

The original paper compares the individual score of agent $p_1$ to the collective score of the group (the mean score of all agents) to track the dynamics of the negotiation throughout the game. Building on this, we propose to measure inequality among the agents to explore how the negotiation process affects the distribution of outcomes. This allows us to observe whether the game produces fair outcomes or if certain agents are disadvantaged—particularly since a successful deal only requires five out of six players to agree. To measure inequality, we use the Gini coefficient (Dorfman, 1979), a commonly used metric that ranges from 0 (perfect equality) to 1 (maximum inequality). We use Equation 1 to compute the Gini coefficient, where $x_i$ and $x_j$ are the gains of agents, $n$ is the number of agents, and $\bar{x}$ is the mean gain.

$$\text{Gini} = \frac{\sum_{i=1}^{n} \sum_{j=1}^{n} |x_i - x_j|}{2n^2 \bar{x}} \tag{1}$$

We report the Gini coefficient for the different game variants (compromising, greedy, untargeted adversarial, and targeted adversarial) and the multi and single-agent setups with CoT configuration of Row 2 from Table 3. This CoT configuration was chosen to give greedy and adversarial agents more opportunity to analyze

| Model | Final (%) ↑ | | Any (%) ↑ | Wrong (%) ↓ | Failed Experiments (%) ↓ | Structure Leakage (%) ↓ |
|---|---|---|---|---|---|---|
| | **5-way** | **6-way** | | | | |
| GPT-4 | 70 | 10 | 100 | 10 | 0 | 10 |
| GPT-4o Mini | 80 | 0 | 100 | 11.15 | 0 | 0 |
| Llama 3.3-70B | 70 | 10 | 100 | 7.3 | 0 | 0 |
| Llama 3.0-70B | 40 | 10 | 90 | 7 | 0 | 0 |
| Qwen 2.5 32B | 60 | 30 | 90 | 9.2 | 0 | 0 |
| Llama 2-13B | - | - | - | - | 80 | - |
| Phi-4 14B | 30 | 0 | 90 | 8.84 | 0 | 0.4 |
| Llama 3.1-8B | 44 | 11 | 88 | 18 | 10 | 38.8 |
| Qwen 2.5 7B | 30 | 10 | 50 | 14.2 | 0 | 45.8 |
| DeepSeek-R1 7B | 0 | 0 | 0 | 48 | 0 | 43.1 |
| Qwen 2.5 1.5B | 30 | 30 | 30 | 19.5 | 60 | 66.5 |

Table 2: Performance of all models using the metrics from Abdelnabi et al. (2024) with 10 experiments per model. We introduced *Failed Experiments* metric counting experiments in which the model failed to complete the negotiation (we describe these problems in Section A.2). The *Structure Leakage* column comes from the leakage comparison metrics discussed in Table 11.

| Model | row no. | CoT: Observation | | CoT: Exploration | | CoT: Planning | Final (%) ↑ | | Any (%) ↑ | Wrong (%) ↓ | Leak (%) | Gini |
|---|---|---|---|---|---|---|---|---|---|---|---|---|
| | | Prev. deals | Others' prefer. | Candidates | Selection | | 5/6-way | 6-way | | | | |
| GPT-4o Mini | 1 | ✗ | ✗ | ✗ | ✗ | ✗ | 30 | 0 | 100 | 8.85 | 0 | 0.15 |
| | 2 | ✓ | ✓ | ✓ | ✓ | ✓ | 70 | 0 | 100 | 2.71 | 0.8 | 0.14 |
| | 3 | ✓ | ✓ | ✗ | ✓ | ✓ | 70 | 0 | 90 | 1.92 | 0 | 0.13 |
| | 4 | ✓ | ✓ | ✗ | ✓ | ✗ | 70 | 0 | 100 | 1.92 | 0 | 0.11 |
| | 5 | ✗ | ✓ | ✗ | ✓ | ✓ | 80 | 0 | 100 | 11.15 | 0 | 0.11 |
| | 6 | ✗ | ✗ | ✗ | ✓ | ✓ | 70 | 0 | 70 | 10.77 | 0 | 0.14 |

Table 3: This table replicates Table 1 from Abdelnabi et al. (2024), replacing GPT-4 with GPT-4o Mini. Elements highlighted in yellow indicate modifications to the specific CoT structures. Our results align with the original paper, showing that skipping calculations for previous deals or candidate proposal generation improves performance.

the negotiation. Due to the lack of communication with other parties and selfish incentives, single-agent, greedy, and adversarial variants will likely lead to higher Gini coefficients than in the compromising game.

# 5 Results

## 5.1 Reproducibility of Original Paper

To reproduce the original paper's results as closely as we could, we evaluated the performance of different models on the original multi-agent negotiation benchmark. Using each of these models to act as different negotiation agents, we compared their ability to produce successful deals and follow the structured negotiation process rules. Table 2 presents the performance of all tested models. The performance of different models was evaluated using the metrics defined in the original benchmark, described in 3.3.

To illustrate the correlation between model size and performance, we refer to Figure 1. It clearly shows that increasing model size generally improves performance. Smaller models, such as Qwen 2.5 1.5B and DeepSeek-R1 7B, struggled to reach acceptable deals and exhibited a high percentage of incorrect deals or failed experiments.

To further evaluate the reproducibility of the original paper, we conducted an ablation study to examine the impact of prompt structure on performance, as shown in Table 3. We systematically tested various CoT structures—such as calculating scores for previous deals, inferring other agents' preferences, generating candidate proposals, and planning—to asses their effect on the negotiation process.

These ablation results are further illustrated in Figure 2, which visualizes the progression of deals from $p_1$'s perspective over multiple rounds for each configuration. In the best-performing configuration (Figure 2a),

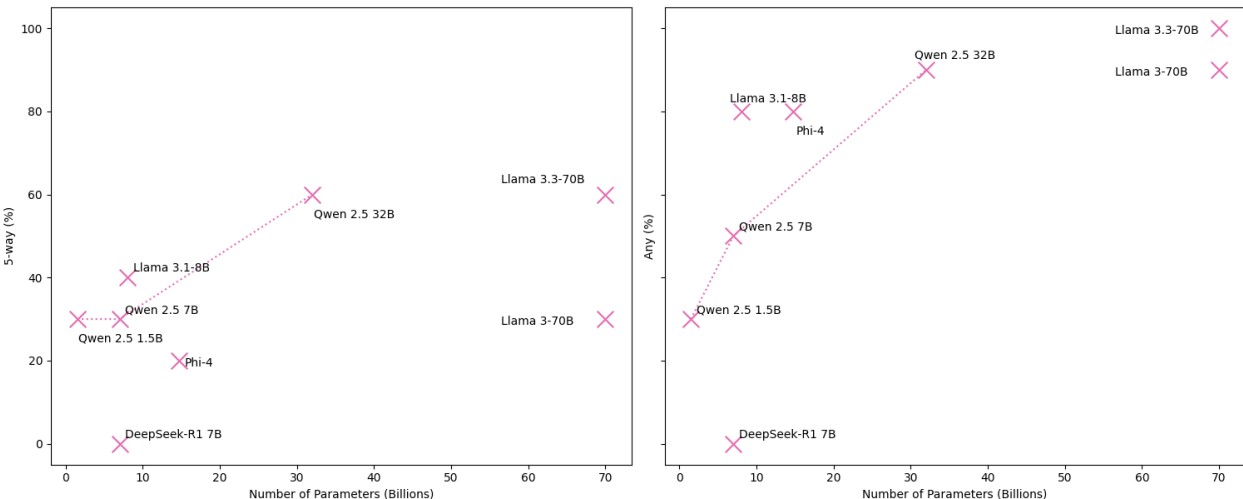

Figure 1: The figure illustrates how performance improves for larger models when controlling for model provider and version (dotted line). It also shows performance gains in newer model versions—e.g., Llama 3.3 70B compared to Llama 3 70B. A more detailed analysis of the failure modes for smaller models and the impact of model size can be found in Section A.2

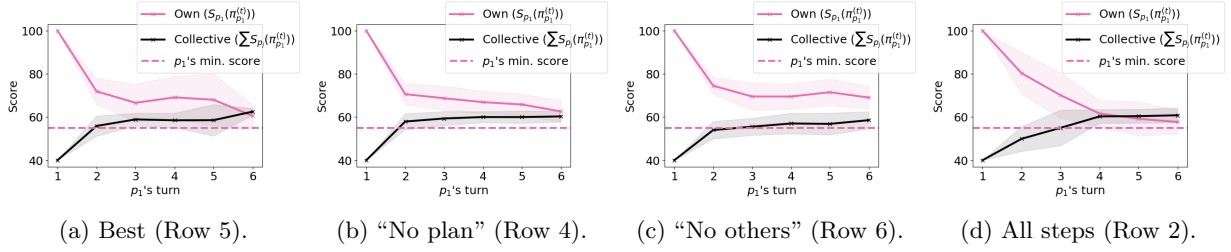

Figure 2: $p_1$'s deals progression over rounds of GPT-4o Mini experiments in Table 3. The results closely match those reported by Abdelnabi et al. (2024) for GPT-4.

Row 5), $p_1$ refines the deal to increase the collective score while decreasing their own, ultimately leading to a successful agreement.

In contrast, configurations that omit essential steps (Figures 2b and 2c, Row 4 and Row 6 respectively) stagnate and saturate on the final rounds, with little improvement in scores and reduced agreement success rates. Meanwhile, the "all steps" configuration (Figure 2d, Row 2) suggests that agents tend to prioritize self-beneficial deals over collective success.

Additionally to the CoT ablation study, we explore different variations of prompts for generalizability purposes on the row 5 CoT combination. We compare the base game with base rewritten, provided by the original paper, where the negotiation scenario is rephrased using GPT-4. We further explore whether the order of information in the prompts matters, by changing the order of prompt sections (individual instructions, scoring rules, voting rules and incentive rules). The results, in Table 10, show that these perturbations do not significantly influence the performance on the benchmark, as tested on GPT-4o Mini, which is favorable for generalizability of the benchmark.

Overall, these results highlight the importance of both model size and CoT prompt structure in successful multi-agent negotiations. Our findings reaffirm the original paper, showing that while larger models tend to perform better, prompt design remains crucial for robust performance. Together, these factors are essential in evaluating the quality and effectiveness of the benchmark itself.

| Game | Acceptable Deals | Pareto Size | Min/Avg/Max Score | Min/Avg/Max Inequality |
|---|---|---|---|---|
| Base | 77/720 | 62/77 | 51.5/60.5/69.8 | 0.04/0.14/0.26 |
| Game 1 | 57/720 | 47/57 | 56.3/67.5/72.3 | 0.05/0.09/0.14 |
| Game 2 | 66/720 | 50/66 | 46.5/57.2/66.3 | 0.03/0.15/0.27 |
| Game 3 | 89/720 | 34/89 | 54.2/70.1/85.7 | 0.03/0.09/0.21 |

Table 4: Results for the Pareto front analysis for all game variations. All unacceptable deals are outside of Pareto front. Across all game runs all suggested acceptable deals are inside the Pareto front. Note, that the game statistics (Acceptable Deals) is different from the one provided by authors, since we found a bug in the code corresponding to game analysis.

## 5.2 Pareto Efficiency Analysis

Table 4 shows that almost all acceptable deals are on the Pareto front while all the unacceptable deals are outside of it. Since all non-acceptable deals result in a score composed of acceptance thresholds (BATNAs), all acceptable deals dominate the non-acceptable ones. Without the BATNA rule, the game's Pareto front would be 705/720.

The game is non-zero-sum, as the authors state because the BATNA rule limits the Pareto front almost entirely to the set of acceptable deals. However, the game becomes almost zero-sum as soon as the models reach the set of acceptable deals. We identify two main implications of this finding:

1. Models have no collective incentive to improve deals since there is no room for optimization beyond acceptability. The problem that the models are collectively solving is not optimization of the score, but identifying any deal from the subset of acceptable deals.

2. Almost all acceptable deals are non-comparable in terms of quality: improving the score of one party means worsening the score of another.

## 5.3 Single-agent Setup

Our results show that the single-agent configuration performs just as well as the original multi-agent setup. Without communicating with other agents, and using only information from the global initial game rules provided, the model can predict acceptable deals with the same success rate as the multi-agent setup.

Figures 3 and 5 compare the single-agent setup to the original multi-agent game using the CoT configuration from Row 5 of Table 3. The results indicate that the single-agent setup closely matches the multi-agent performance and, in some cases, even outperforms it in terms of collective scores. Notably, both models achieve very similar performance between the two setups.

Table 5 further supports this trend, showing that the final deals proposed in the single-agent setup are comparable to–or even better than—those using the multi-agent configuration. These results challenge the original claim that the benchmark evaluates cooperation, communication, and negotiation skills, as they suggest that agent interaction is not essential for success. The same pattern emerges across different game configurations, as shown in Figure 6, suggesting that the benchmark does not effectively evaluate the communication and negotiation skills of the agents.

The model that performs best in the single-agent setup is GPT-4o Mini, which maintains or improves its 5-way metric compared to the multi-agent scenario. GPT-4's results show that its performance decreases significantly when facing the single-agent scenario. We provide a detailed analysis of this behavior in Section A.3, where we show that the cause is GPT-4 underestimating the needs of P2 or its veto power but accurately estimating the needs for other parties.

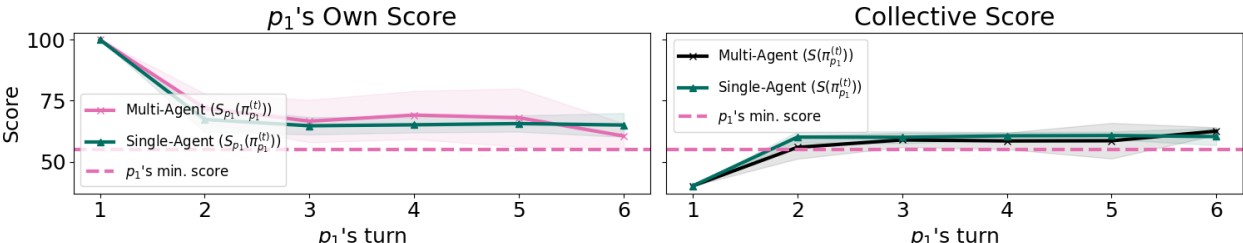

Figure 3: Comparison of the single- and multi-agent approaches in the base negotiation game for GPT-4o Mini. The plots show similar results when we allow communication (pink) vs. no communication (green).

| Model | Configuration | Final (%) ↑ | | Any (%) ↑ | Wrong (%) ↓ | Leak (%) | Gini |
|---|---|---|---|---|---|---|---|
| | | 5-way | 6-way | | | | |
| | Multi-agent | 70 | 10 | 100 | 10 | 10 | 0.12 |
| GPT-4 | Single-agent (6 calls) | 10 | 10 | 50 | 7.13 | 18.33 | 0.14 |
| | Single-agent (1 call) | 10 | 0 | 10 | 0 | 3.33 | 0.20 |
| | Multi-agent | 80 | 0 | 100 | 11.2 | 0 | 0.11 |
| GPT-4o Mini | Single-agent (6 calls) | 80 | 0 | 100 | 0 | 0 | 0.13 |
| | Single-agent (1 call) | 90 | 0 | 90 | 0 | 0 | 0.11 |
| | Multi-agent | 60 | 30 | 90 | 9.2 | 0 | 0.12 |
| Qwen 2.5 32B | Single-agent (6 calls) | Yes | No | Yes | 0 | 0 | 0.11 |
| | Single-agent (1 call) | Yes | No | Yes | 0 | 0 | 0.13 |

Table 5: Comparison of different LLMs and their performance across setups. Qwen's results are interpreted as binary since its output is deterministic in the single-agent variant. Yes and No corresponding to success in the given metric.

## 5.4 What the Game Actually Measures

The game is originally presented as an assessment of how well six agents negotiate the best possible deal. However, as demonstrated above, the problem effectively reduces to finding any acceptable deal—most of which already lie on the Pareto front. A crucial yet unstated detail in the original paper is that the agents do not actually vote themselves. Instead, voting is automated: an agent accepts a deal if it meets its minimum score threshold. This creates an inherent imbalance—only $p_1$ has decision-making power, while the other agents merely provide information. Moreover, our single-agent baseline performs as well as the multi-agent setting, suggesting that $p_1$ generates equally strong deals regardless of communication. This implies that information from the other agents is either redundant or not utilized by $p_1$.

Another experiment that further supports our claim is presented in Table 6. This table illustrates how different assignments of veto power between the regular player and the adversarial agent affect the game's score. The most surprising results appear in scenarios where the adversarial agent holds veto power over the game's outcome (first two lines). While one might expect this combination of veto power and a disruptive objective to heavily damage the score, our experiments show no significant change in the score. Other lines show a similar pattern of no notable change. We conclude that veto power dynamics among the group members, excluding $p_1$, have little to no impact on the final outcome of the game, highlighting the game's heavy imbalance towards $p_1$.

In summary, the game reduces to all agents stating their preferences while $p_1$ selects an acceptable deal—something it can accomplish effectively using only the predefined rules. We hypothesize that the game primarily measures how well a model infers a compromise based on the setup and adheres to the formatting. These findings contradict the claim that the benchmark evaluates cooperation, communication, and negotiation skills.

| Model | Veto owner | Adversarial Agent | Final (%) ↑ | | Any (%) ↑ | Wrong (%) ↓ | Gini |
|---|---|---|---|---|---|---|---|
| | | | 5-way | 6-way | | | |
| | Department of Tourism | Department of Tourism | 80 | 0 | 100 | 0.38 | 0.13 |
| | Environmental activists | Environmental activists | 60 | 0 | 70 | 0 | 0.14 |
| GPT-4o Mini | Department of Tourism | - | 70 | 0 | 100 | 2.71 | 0.14 |
| | Environmental activists | - | 80 | 20 | 80 | 3.42 | 0.11 |
| | Department of Tourism | Environmental activists | 90 | 30 | 90 | 0 | 0.12 |
| | Environmental activists | Department of Tourism | 80 | 10 | 90 | 12.7 | 0.12 |

Table 6: Comparison of veto power dynamics in the untargeted adversarial setting. We run the experiments on the base game with row 2 configuration, where one agent is untargeted adversarial, two agents have veto power (one of them being $p_1$). Note that in all other tables within this and the reproduced paper, the Department of Tourism is always $p_2$, and Environmental activists are always adversarial in all adversarial variations.

## 5.5 Proposed New Metrics

### 5.5.1 Structure Leakage

The single-agent baseline has shown that the rules of the game and agent descriptions already reveal some information to the players. This leads us to another source of information leaking: *Structural Leakage*. This metric shows leakage of confidential information due to LLMs' incorrect formatting of outputs. The implementation decision to rely on LLMs to correctly format their answers is detrimental to the benchmark's effort for a fair comparison of models. As our results show, small models had a harder time strictly adhering to formatting requirements, already disadvantaging them in negotiation under this implementation. As shown in Table 11, structural leakage is directly correlated with the model size—larger, better-performing models have no structural leakage, while the smaller models experience substantial leakage.

We initially expected the original leakage metric to always be higher than structural leakage due to its presumed broader scope. The original metric uses an LLM as a judge, which should be capable of detecting both explicit structural errors (e.g. missing plan or scratchpad tags) and more subtle forms of leakage. However, this is not always the case since small Qwen models seem to perform better on the original leakage metric. The structure leakage detects leakage of plans or scratchpads by identifying tags such as `<PLAN>`, but the original metric does not consider these cases as leakage because agents do not reveal information about confidential scores. We argue that this behavior is a form of leakage because it disrupts the negotiation dynamics.

To ensure all types of leakage are properly detected, a more robust prompt for the LLM judge is needed. In addition, we have computed the intersection of the two metrics to show how much leakage could be potentially prevented if different implementation choices were made, see Table 11. For small models, it is apparent that most of the leakage occurs because entire answers are revealed to others due to a lack of tag adherence. We argue that the benchmark should handle the different CoT steps separately to avoid structural leakage and not hinder smaller models because of formatting problems.

### 5.5.2 Inequality

Results from Figure 7 show that the highest inequality is reached in the greedy game variants, which is expected since most of the proposed deals are not acceptable. Secondly, we showcase that in the single-agent setup, deals are slightly more unequal since $p_1$ is the only one proposing deals without others' input. Surprisingly, we show that adversarial games lead to deals with inequality similar to compromising games. However, from the single-agent baseline in Figures 3 and 5, we see that $p_1$ does not take much input from other agents. Therefore, it is likely that the adversarial agent does not considerably change the dynamic of the game, explaining similar inequality scores of the deals. This is further confirmed by Table 6, where we show the influence of different veto power configurations on the outcome of the game. The inequality metric stays roughly the same independent of either the adversarial agent has veto power, which further proves our point of outcome of the game relying predminantly on $p_1$. However, Table 5 shows that using

different prompt ablations yields different results. Therefore, each game variant can prefer a different CoT configuration for its best performance. We argue that reporting the inequality enhances the benchmark by capturing variations in individual scores, providing a more comprehensive evaluation of agent performance.

# 6 Discussion

Our study aimed to replicate the main claims of the paper "Cooperation, Competition, and Maliciousness" (Abdelnabi et al., 2024), which proposes a benchmark for LLM negotiation. Furthermore, we propose several extensions to test the claims and advance the focus on fairness and confidentiality.

Our main result shows that the benchmark fails to measure the communication and negotiation between agents, disproving the core claim of the original paper, Claim 5, about the importance of these LLM skills for successful negotiation in the benchmark's game. We demonstrate this using an equally well-performing baseline where no communication is present. Secondly, our game analysis shows that the game is technically non-zero-sum as the original paper claims in Claim 4. However, once agents reach the set of acceptable deals, further improvement is not possible. This indicates that the benchmark prioritizes finding sufficiently good deals rather than necessarily identifying the optimal one (generating the highest optimal score). We show that Claim 2 holds since large models outperform most small models with the exception of GPT-4o Mini, whose size is, however, not explicitly disclosed. Similarly, we support Claim 1, as we show that open-weight models fall behind their SOTA closed-weight counterparts. Finally, we confirm that CoT steps are essential for each agent to reach their own and collective goals, Claim 3. The results of the extensions to the original work show that:

1. The game is not about negotiating the best deal, but $p_1$ finding any acceptable deal.

2. An agent can propose a successful deal without communicating with the other parties.

3. The proposed leakage metric addresses gaps in the original approach for detecting leakage and their implementation decisions for handling LLM outputs.

4. The inequality metric helps to identify the presence of greedy agents in the negotiation.

5. Finally, the main failure modes of small models are self-monologues and answer formatting.

## 6.1 Limitations and Future Work

A key limitation of this study is the sensitivity of results to prompt variations. The CoT ablation of prompts from the original paper was tested only on GPT-3.5 and GPT-4. From our results from Table 5 and 8, we can see that different prompt configurations yield various results for different models. This is a big limitation of the benchmark as well as of our results. This dependency suggests that results may reflect an agent's responsiveness to specific prompt formulations rather than its underlying negotiation abilities. Additionally, it raises concerns about reproducibility, as minor prompt variations could lead to significantly different outcomes among the models tested.

Secondly, due to constraints on computational resources, we did not perform all of the experiments on models larger than 32B parameters. Therefore, there might be other open-weight models outperforming SOTA closed-weight models. We focused our efforts on benchmarking GPT-4o Mini and GPT-4 but acknowledge that models like GPT-4o are also relevant for evaluation because they provide a good tradeoff between cost and performance. Future improvements in LLM negotiation benchmarks should develop a more robust approach to prompt crafting. There is also room for improvement in the game design itself. We suggest designing a similar benchmark but with a non-zero-sum set of acceptable deals, a smaller Pareto front, and non-automated voting among agents. Finally, to ensure the benchmark tests the communication and negotiation skills of agents, our communication-less baseline should not be competitive.

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

# A   Appendix

## A.1   Example of Information in Game Description

We provide an example of how the game's description can provide $p_1$ the information required to achieve a successful deal. In the base game designed by Abdelnabi et al. (2024), issue A (*Infrastructure Mix*), related to whether the facilities will be built on land or water, the *Environmental League* already knows that $p_1$, named *SportCo*, will push for solution A1 (water-based, the least restrictive option for *SportCo*), and oppose A3 (land-based, the most restrictive). Similarly, in issue B (*Ecological Impact*), $p_1$ can infer that the *Environmental League* will strongly favor B3 (environmental improvement) and likely reject B1 (permanent damage), making the negotiation unnecessary. Finally, in issue E (*Compensation to other cities*), it is clear that *Other cities* will push for a higher compensation, making E5 (no compensation) unacceptable. As a result, $p_1$ can immediately propose a compromise, such as E3 (paying $300 million) instead of E1 (paying $600 million) without further negotiation.

Additionally, issues B and E provide explicit context before listing the available options for the agents. For example, issue B states '*The "Environmental League" argues that this project might damage local dolphins and sea lion populations.*', while issue E says that '*"Other cities" in the area believe their local tourism will be harmed by this project and therefore they are requesting compensations.*". These descriptions further reinforce the predictability of each agent's preference.

Providing this information to all agents, especially $p_1$ who has the final say in the proposal, reduces the necessity of direct communication between the agents for successfully reaching an agreement. To evaluate this, we test a scenario where $p_1$ generates deal proposals without input from other parties. By relying only on the provided information, its reasoning skills, and planning capabilities, we assess whether it can independently come up with an agreeable deal that satisfies all parties.

## A.2   Impact of Model Size

To ensure structured communication, the benchmark requires all the agents to format their responses according to a predefined structure provided through the CoT prompts in each round. For example, agents must enclose their proposals for the other agents within `<ANSWER>` tags (commonly referred to as the *public answer*), while their private reasoning must be enclosed within `<SCRATCHPAD>` tags (the *private answer*). Additionally, any deal proposal must be enclosed within `<DEAL>` tags. Properly using these tags is crucial to

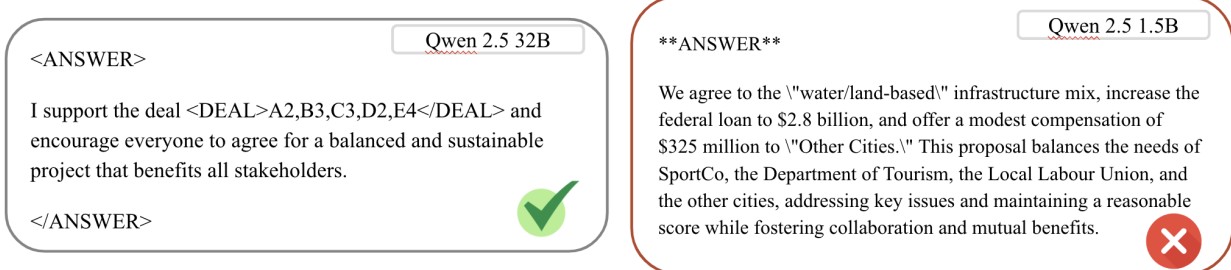

Figure 4: Example final answer of $p_1$ agent using Qwen 2.5 32B (left) and Qwen 2.5 1.5B (right). The answer on the left shows how the agent fails to include a deal enclosed in `<ANSWER>` and `</ANSWER>` with specific mentions of the deal that resulted from the negotiation. Such answers led to the experiments being classified as failed because we could not parse the outcome of the negotiation.

evaluate how well a model follows instructions and adheres to the benchmark's structure, as failure to do so may result in an inability to assess its capabilities.

The models with less than 8B parameters tested by Abdelnabi et al. (2024) were not included in their results because they did not follow the game's rules. However, the authors do not provide details of their failure modes. Our results from small model agents show that the two main reasons for failure are failing to include the correct tags in the answer and engaging in irrelevant self-monologues.

As shown by Table 2, we could not parse deals from Qwen 2.5 1.5B in 60% of the experiments. The reason for the failed experiments was responses such as Figure 4. The agent answered with related topics but used `**ANSWER**` instead of the `<ANSWER>` and `</ANSWER>` tags to enclose the `public answer`. Additionally, the agent does not include the `<DEAL>` tag with the deal proposal to show the others. We also encountered instances with hallucinated tags like `<SUGGESTION>`. These results align with Abdelnabi et al. (2024), as the small models did not follow the game instructions correctly. Another reason for failure was engaging in irrelevant self-monologues or responding with random words and characters that did not add value to the negotiation process. This was the cause of the failed experiments for Llama 3.1 8B and Llama 2 13B, shown in Table 2. Consequently, this increased the required computing resources because it made the conversation history longer.

Under our definition of small models, GPT-4o Mini is in the category of small models according to the 8B parameter count estimated by Abacha et al. (2024). This would then disprove the claim that small models were not able to perform well on the benchmark since GPT-4o Mini was the best model we tested. The number of parameters of closed-source models is uncertain, thus, we restrict our analysis of the author's claim to open-weight models. We also analyzed how model performance changed when we used larger models for the negotiation. The results are summarized in Figure 1, where we show that larger models–i.e., models of 70B parameters, perform better than other models we used. The trend is more noticeable when we compare models controlling for the model provider and version as we do with the Qwen 2.5 model family. Additionally, each model version may incorporate different architectural or data processing advancements that impact the negotiation outcome. For example, Figure 1 shows that Llama 3.3 70B has an improvement of 100% in the 5-way metric and 11% in the Any metric when compared to Llama 3 70B.

### A.3 Analysis of GPT-4 on the Single-Agent Baseline

We looked into the single-agent's performance for GPT-4, and the main conclusion is that GPT-4 is under-estimating the needs of P2 or its veto power but accurately estimating the needs for other parties. P2 rejects the final deal with its veto power, and that is why the deals are unsuccessful. Compared to the multi-agent scenario, P2 cannot provide additional information to P1 during the interaction and help calibrate its needs to reach a successful deal in the single-agent setup. Without the veto power of P2, most of the deals would be accepted as 5-way deals.

| Model | Single Agent (6 rounds) | Single Agent (1 round) |
|---|---|---|
| **GPT-4** | A2, B3, C2, D1, E3 | A2, B2, C3, D2, E5 |
| | A2, B2, C3, D1, E2 | A1, B2, C3, D1, E4 |
| | A2, B3, C2, D1, E2 | A2, B2, C3, D1, E5 |
| | A2, B2, C3, D1, E4 | A2, B2, C3, D1, E4 |
| | A2, B2, C3, D1, E4 | A2, B2, C3, D1, E4 |
| | A3, B3, C2, D1, E1 | A2, B2, C3, D1, E4 |
| | A2, B2, C3, D1, E4 | A1, B2, C3, D1, E4 |
| | A2, B2, C2, D1, E2 | A2, B2, C2, D1, E3 |
| | A2, B3, C3, D2, E4 | A2, B2, C3, D2, E3 |
| | A3, B2, C3, D1, E3 | A2, B2, C3, D1, E4 |
| **GPT-4o Mini** | A1, B2, C2, D2, E3 | A1, B2, C3, D2, E4 |
| | A2, B2, C2, D1, E3 | A2, B2, C3, D2, E3 |
| | A2, B2, C2, D2, E4 | A2, B2, C3, D2, E3 |
| | A2, B2, C3, D2, E2 | A2, B2, C3, D2, E3 |
| | A2, B2, C3, D2, E3 | A2, B2, C3, D2, E3 |
| | A2, B2, C3, D2, E2 | A2, B2, C3, D2, E4 |
| | A1, B2, C3, D2, E4 | A2, B2, C3, D2, E3 |
| | A2, B2, C2, D1, E3 | A2, B2, C3, D2, E3 |
| | A2, B2, C3, D2, E3 | A2, B2, C3, D2, E3 |
| | A2, B2, C3, D2, E4 | A2, B2, C3, D2, E4 |

Table 7: Negotiation outcomes for each model-scenario combination in 10 runs. The difference between the two columns is how many refinements P1 can make to the deal because P1 is not communicating with other parties. The main conclusion is that GPT-4 proposes more deals with D1 while GPT-4o Mini proposes more deals with D2. This difference impacts the negotiation because D2 is a key issue for P2.

The results in Table 7 compare the outcome of the negotiation for GPT-4 and GPT-4o Mini for the single-agent setup. In the GPT-4 single-agent game, there is a lot more D1 in the final deals, which lowers the score for P2, who holds the veto, but is more favourable for P1. This happens less for GPT-4o Mini, as can be seen from the results below, showing that GPT-4 underestimates the veto power rule of P2.

### A.4 Computational requirements and energy consumption

We conducted our experiments using a combination of local machines and high-performance computational resources. For proprietary models like GPT-4o Mini, we accessed the model via the OpenAI API, which did not require additional computational resources on our end. For the open-source models, we used the Netherlands' national supercomputer, Snellius, with access to NVIDIA A100 and H100 GPUs.

To monitor energy consumption, we used EAR (Corbalan et al., 2020). Table 1 reports approximately how much energy was consumed per game for each model. The table shows that, in most cases, energy consumption is correlated to the size of the model, with smaller models consuming 20-200 kJ, and larger models using 600-800 kJ.

One notable exception to this is Llama2 13B, which had an unusually high energy consumption due to its excessive self-monologing. The model continuously produced irrelevant text until we stopped it manually. The higher energy consumption of Phi4 and Llama3.1-8B compared to Qwen 32B, although more than two and four times smaller in size, was caused by longer responses from the agents.

Overall, Qwen 32B, the model used in most of the experiments, achieves high scores in the game while using energy amounts comparable to those of smaller models. This made it an efficient choice for large-scale evaluations.

## A.5 Additional Results for Single-Agent Baseline

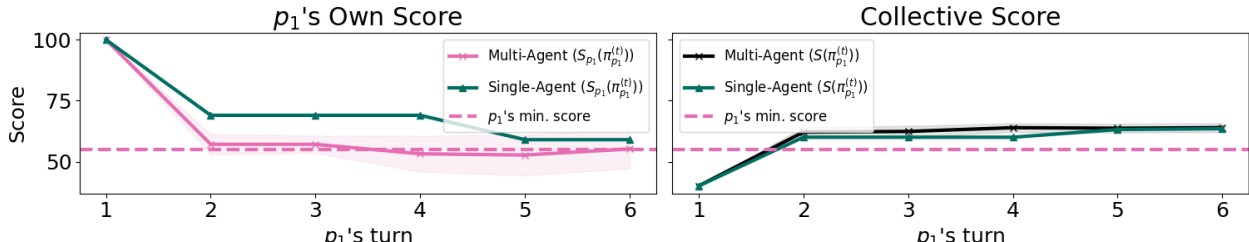

Figure 5: Comparison between the single- and multi-agent approaches in the base negotiation game for Qwen 2.5 32B. The plots show similar results when we allow communication (pink) compared to no communication (green).

| Model | Game Type | Setup | Final (%) ↑ | | Any (%) ↑ | Wrong (%) ↓ | Leak (%) | Gini |
|---|---|---|---|---|---|---|---|---|
| | | | 5-way | 6-way | | | | |
| GPT-4o Mini | Compromising | Multi-Agent | 70 | 0 | 100 | 2.71 | 0.8 | 0.14 |
| | | Single-Agent (6) | 30 | 10 | 90 | 2.85 | 1.67 | 0.19 |
| | Greedy | Multi-Agent | 0 | 0 | 10 | 2.69 | 0 | 0.27 |
| | | Single-Agent (6) | 0 | 0 | 0 | 0 | 0 | 0.42 |
| | Untargeted Adv | Multi-Agent | 90 | 30 | 90 | 0 | 0 | 0.12 |
| | Targeted Adv | Multi-Agent | 70 | 10 | 100 | 0.8 | 0 | 0.13 |
| Qwen 32B | Compromising | Multi-Agent | 100 | 10 | 100 | 0.3 | 0 | 0.12 |
| | | Single-Agent (6) | 100 | 0 | 100 | 0 | 0 | 0.13 |
| | Greedy | Multi-Agent | 30 | 20 | 50 | 2.3 | 0 | 0.19 |
| | | Single-Agent (6) | 0 | 0 | 0 | 0 | 0 | 0.20 |
| | Untargeted Adv | Multi-Agent | 100 | 10 | 100 | 0 | 0 | 0.12 |
| | Targeted Adv | Multi-Agent | 90 | 10 | 100 | 0 | 0 | 0.13 |

Table 8: Gini inequality values of final deals in Multi-agent and Single-Agent setup, across different game variants with Row 2 prompt configuration.

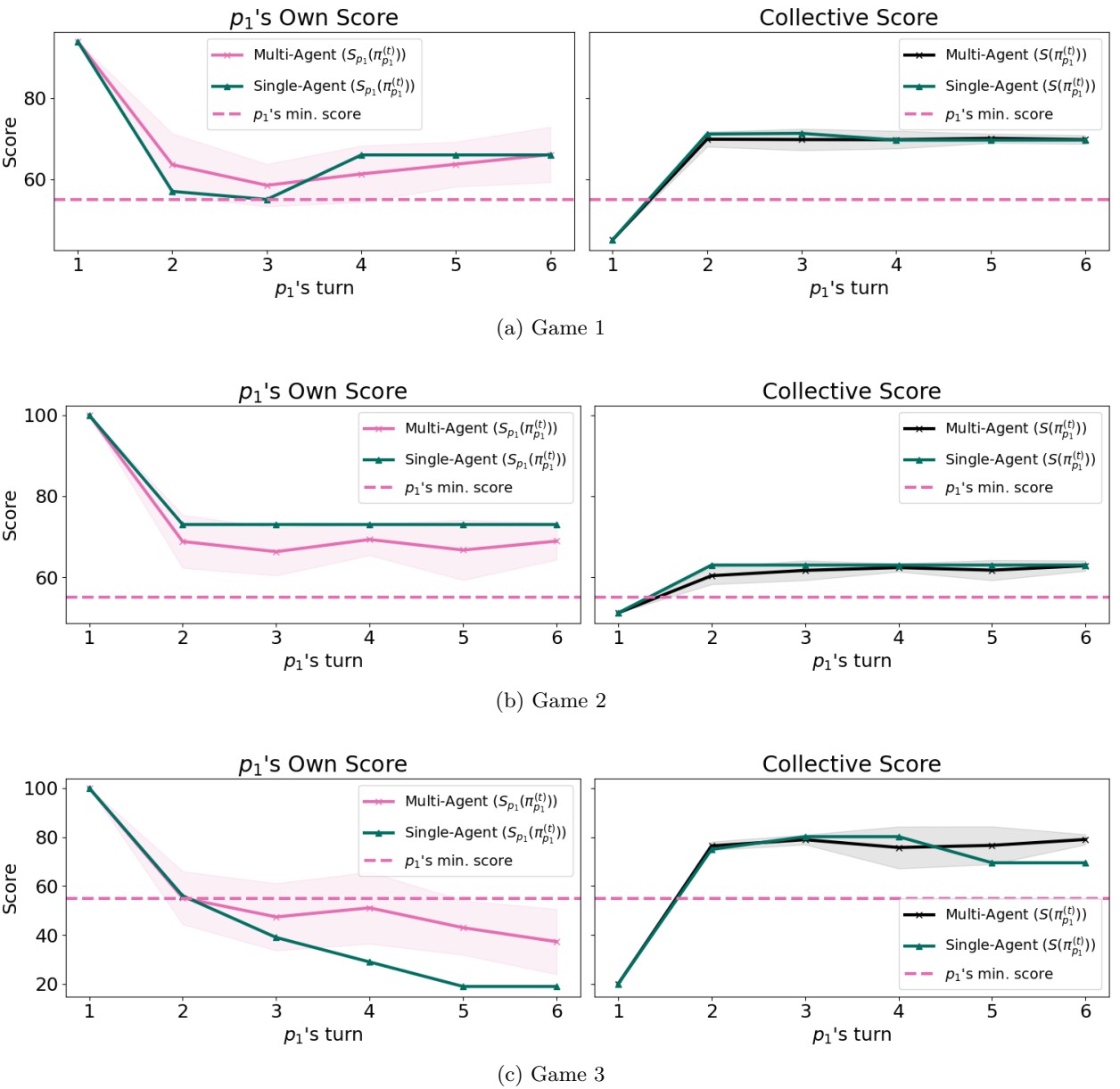

Figure 6: Results for games 1, 2, and 3 for Qwen 2.5 32B. The plots show similar behavior for the base game in Figures 5, where the single-agent setup has a similar result to the multi-agent setup.

## A.6  Additional Results for Game Variants

| Model | Game | Final (%) ↑ | | Any (%) ↑ | Wrong (%) ↓ | Leak (%) | Gini |
|---|---|---|---|---|---|---|---|
| | | 5-way | 6-way | | | | |
| GPT-4o Mini | Base | 80 | 0 | 100 | 11.2 | 0 | 0.11 |
| | Game 1 | 40 | 20 | 100 | 0.38 | 0 | 0.10 |
| | Game 2 | 0 | 0 | 10 | 10 | 0 | 0.10 |
| | Game 3 | 10 | 0 | 90 | 10.38 | 0 | 0.12 |
| Qwen 2.5 32B | Base | 60 | 30 | 90 | 9.2 | 0 | 0.12 |
| | Game 1 | 30 | 20 | 90 | 10.2 | 0 | 0.09 |
| | Game 2 | 10 | 0 | 40 | 11.88 | 0 | 0.11 |
| | Game 3 | 20 | 0 | 90 | 11.53 | 0 | 0.14 |

Table 9: Comparison of performance of models across the different game setups.

## A.7  Different prompt variations of the game instructions

| Model | Final (%) ↑ | | Any (%) ↑ | Wrong (%) ↓ |
|---|---|---|---|---|
| | 5-way | 6-way | | |
| Base | 80 | 0 | 100 | 11.15 |
| Base rewritten | 70 | 10 | 100 | 11.54 |
| Base rewritten perturbed | 80 | 0 | 100 | 11.15 |

Table 10: Performance of GPT-4o Mini on different prompt variations of the base game. The first row is the same configuration in Table 2, the second row is the rewritten base game used by Abdelnabi et al. (2024) in the original paper, and the third row is the same game as row 1 but changing the order of the prompt sections.

## A.8 Detailed Results for Additional Metrics

| Model | Structure Leakage (%) ↓ | Original Leakage (%) ↓ | Intersection(%) |
|---|---|---|---|
| GPT-4o Mini | 0 | 0 | 0 |
| Llama 3.3-70B | 0 | 0 | 0 |
| Llama 3.0-70B | 0 | 4.23 | 0 |
| Qwen 2.5 32B | 0 | 2.31 | 0 |
| Llama 2-13B | - | - | - |
| Phi-4 14B | 0.4 | 5.38 | 0 |
| Llama 3.1-8B | 38.8 | 51.75 | 25 |
| Qwen 2.5 7B | 45.8 | 7.31 | 5.77 |
| DeepSeek-R1 7B | 43.1 | 63.08 | 39.62 |
| Qwen 2.5 1.5B | 66.5 | 31.54 | 22.69 |

Table 11: Structure and Original Leakage Comparison

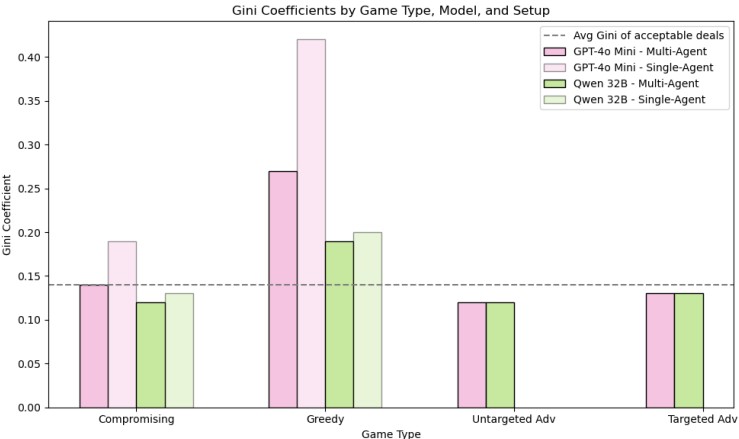

Figure 7: Gini coefficients of inequality for the different game variants and single/multi agent setups, for Row 2 prompt configuration

### A.9 Additions to the author's code

- Originally, the authors set the model temperature to 0 and a random order of the agents for each round. However, the authors did not add a fixed seed in the code, making their exact results not reproducible. To address this, we added fixed seeds to the code setup. We then ran all the experiments 10 times with seeds ranging from 1 to 10. We also corrected the `do_sample` parameter in the Hugging Face pipeline, changing it from `True` to `False` to set up the correct greedy decoding configuration.

  Although the original paper performed 20 experiments, we reduced the number of experiments due to computational constraints. This decision was also influenced by the fact that the exact reproduction of the results is not possible.

- The authors' code uses fixed `float32` precision for models loaded from Hugging Face, which can be inefficient and affect the generalization for models like Phi-4. We resolved this issue by using `auto` precision, which allows the framework to automatically select the most appropriate precision for the available hardware and model requirements. Additionally, we manually adjusted the precision of Llama models to `float16` precision. This configuration was recommended for managing memory constraints while maintaining an acceptable performance. It is important to note that `float16` is not a form of quantization but a more efficient memory format.

- In their prompts, authors ask models to format their answers using specific tags, such as `<DEAL>`, `<PLAN>`, `<SCRATCHPAD>`, and `<ANSWER>`. These tags are later used to break the model's answer into the Chain-of-Thought structure. Models generally comply with these requirements. However, during evaluation, the `<DEAL>` tags are not used for parsing; instead, the suggested deals are parsed directly from the models' public answers. This approach often leads to unparsable deals, negatively impacting the scores of certain models.

  We consider this a bug because models sometimes respond in a more human-like manner, describing the deals' meanings (e.g., "a loan of 200 dollars" instead of "option A2"). Additionally, the prompts are misleading, implying that the placement of the `<DEAL>` tag within the answer does not matter. This inconsistency skews the evaluation scores. We have addressed this issue and now report the corrected scores.

- We found a bug in the game evaluation: the number of acceptable deals was computed incorrectly, leading to incorrect levels of difficulties and game analysis in the 'Turning the game difficulty' section of the original paper.

- We found a bug in the evaluation notebook: the code inconsistently evaluated if the deal was acceptable. In one part of the code, the score was required to be greater than the minimal threshold, but in the rest of the code, a score equal to the minimal threshold was considered acceptable. After fixing the bug few model evaluations changed.

- We also fixed minor bugs, such as bracket mismatch, setting the value of the API key, faulty conditional parses, and missing the `matplotlib` dependency.

### A.10 Cooperative prompts structures

---

The following is a chronological history of up to `[WINDOW SIZE]` interactions `<HISTORY>` `[HISTORY]` `</HISTORY>`. Now it is your turn to talk.

Please use a scratchpad to explain yourself and why you are agreeing with a deal or suggesting a new one. In your scratchpad,

1) Based on the history and your notes, make a proposal that balances between your scores and accommodating others and that is more likely to lead to an agreement.

You must follow these important negotiation guidelines in all your suggestions:

- Aim for a balanced agreement considering all parties' interests.
- Show flexibility and openness to accommodate others' preferences.
- Express your objectives clearly and actively listen to others.
- Empathize with other parties' concerns to foster rapport.
- Focus on common interests to create a win-win situation.
- It is very important for you that you all reach an agreement as long as your minimum score is met.

Enclose the scratchpad between `<SCRATCHPAD>` and `</SCRATCHPAD>`. The scratchpad is secret and not seen by other parties. Your final answer is public and must never contain scores. Enclose your final answer after the scratchpad between `<ANSWER>` and `</ANSWER>`. Make your final answer very short and brief in 2-3 sentences and containing only your main proposals. Use options' short notations instead of long descriptions. Enclose any deals you suggest between: `<DEAL>` `</DEAL>`.

---

Figure 8: Cooperative prompts structures - 1st Variation: No CoT

The following is a chronological history of up to `[WINDOW SIZE]` interactions `<HISTORY> [HISTORY]` `</HISTORY>`. The following are your previous plans from last interactions. You should follow them while also adjusting them according to new observations. `<PREV PLAN> [LAST PLAN] </PREV PLAN>`. Now it is your turn to talk.

Please use a scratchpad to show intermediate calculations and explain yourself and why you are agreeing with a deal or suggesting a new one. You should map the individual options to their scores denoted by the number between parentheses. You have a calculator tool at your disposal, where you simply add scores of the options to determine the total score of a deal. In your scratchpad,

1) Think about the others' preferences.

2) Based on the others' preferences, history and your notes, make three different proposals that balance between your scores and accommodating others and that is more likely to lead to an agreement.

3) From the three proposals, select the one final deal that is more likely to lead to an agreement.

You must follow these important negotiation guidelines in all your suggestions:

- Aim for a balanced agreement considering all parties' interests.
- Show flexibility and openness to accommodate others' preferences.
- Express your objectives clearly and actively listen to others.
- Empathize with other parties' concerns to foster rapport.
- Focus on common interests to create a win-win situation.
- It is very important for you that you all reach an agreement as long as your minimum score is met.

Enclose the scratchpad between `<SCRATCHPAD>` and `</SCRATCHPAD>`. The scratchpad is secret and not seen by other parties. Your final answer is public and must never contain scores. Enclose your final answer after the scratchpad between `<ANSWER>` and `</ANSWER>`. Make your final answer very short and brief in 2-3 sentences and containing only your main proposals. Use options' short notations instead of long descriptions. Enclose any deals you suggest between: `<DEAL> </DEAL>`.

After the final answer, building on your current move and analysis, briefly write down short notes for yourself of what exact options you can explore the next time you speak. Enclose the notes between `<PLAN>` and `</PLAN>`.

Figure 9: Cooperative prompts structures - 2nd Variation: CoT - Complete

The following is a chronological history of up to [WINDOW SIZE] interactions <HISTORY> [HISTORY] </HISTORY>. The following are your previous plans from last interactions. You should follow them while also adjusting them according to new observations. <PREV PLAN> [LAST PLAN] </PREV PLAN>. Now it is your turn to talk.

Please use a scratchpad to show intermediate calculations and explain yourself and why you are agreeing with a deal or suggesting a new one. You should map the individual options to their scores denoted by the number between parentheses. You have a calculator tool at your disposal, where you simply add scores of the options to determine the total score of a deal. In your scratchpad,

1) Think about the others' preferences.

2) Based on the others' preferences, history and your notes, make a proposal that balances between your scores and accommodating others and that is more likely to lead to an agreement.

You must follow these important negotiation guidelines in all your suggestions:

- Aim for a balanced agreement considering all parties' interests.

- Show flexibility and openness to accommodate others' preferences.

- Express your objectives clearly and actively listen to others.

- Empathize with other parties' concerns to foster rapport.

- Focus on common interests to create a win-win situation.

- It is very important for you that you all reach an agreement as long as your minimum score is met.

Enclose the scratchpad between <SCRATCHPAD> and </SCRATCHPAD>. The scratchpad is secret and not seen by other parties. Your final answer is public and must never contain scores. Enclose your final answer after the scratchpad between <ANSWER> and </ANSWER>. Make your final answer very short and brief in 2-3 sentences and containing only your main proposals. Use options' short notations instead of long descriptions. Enclose any deals you suggest between: <DEAL> </DEAL>.

After the final answer, building on your current move and analysis, briefly write down short notes for yourself of what exact options you can explore the next time you speak. Enclose the notes between <PLAN> and </PLAN>.

Figure 10: Cooperative prompts structures - 3rd Variation: CoT - No Candidate Generation

The following is a chronological history of up to `[WINDOW SIZE]` interactions `<HISTORY>` `[HISTORY]` `</HISTORY>`. Now it is your turn to talk.

Please use a scratchpad to show intermediate calculations and explain yourself and why you are agreeing with a deal or suggesting a new one. You should map the individual options to their scores denoted by the number between parentheses. You have a calculator tool at your disposal, where you simply add scores of the options to determine the total score of a deal. In your scratchpad,

1) Think about the others' preferences.

2) Based on the others' preferences, history and your notes, make a proposal that balances between your scores and accommodating others and that is more likely to lead to an agreement.

You must follow these important negotiation guidelines in all your suggestions:

- Aim for a balanced agreement considering all parties' interests.
- Show flexibility and openness to accommodate others' preferences.
- Express your objectives clearly and actively listen to others.
- Empathize with other parties' concerns to foster rapport.
- Focus on common interests to create a win-win situation.
- It is very important for you that you all reach an agreement as long as your minimum score is met.

Enclose the scratchpad between `<SCRATCHPAD>` and `</SCRATCHPAD>`. The scratchpad is secret and not seen by other parties. Your final answer is public and must never contain scores. Enclose your final answer after the scratchpad between `<ANSWER>` and `</ANSWER>`. Make your final answer very short and brief in 2-3 sentences and containing only your main proposals. Use options' short notations instead of long descriptions. Enclose any deals you suggest between: `<DEAL>` `</DEAL>`.

Figure 11: Cooperative prompts structures - 4th Variation: CoT - No Candidate Generation nor Planning

The following is a chronological history of up to `[WINDOW SIZE]` interactions `<HISTORY>` `[HISTORY]` `</HISTORY>`. The following are your previous plans from last interactions. You should follow them while also adjusting them according to new observations. `<PREV PLAN>` `[LAST PLAN]` `</PREV PLAN>`. Now it is your turn to talk.

Please use a scratchpad to explain yourself and why you are agreeing with a deal or suggesting a new one. In your scratchpad,

1) Think about the others' preferences.

2) Based on the others' preferences, history and your notes, make a proposal that balances between your scores and accommodating others and that is more likely to lead to an agreement.

You must follow these important negotiation guidelines in all your suggestions:

- Aim for a balanced agreement considering all parties' interests.
- Show flexibility and openness to accommodate others' preferences.
- Express your objectives clearly and actively listen to others.
- Empathize with other parties' concerns to foster rapport.
- Focus on common interests to create a win-win situation.
- It is very important for you that you all reach an agreement as long as your minimum score is met.

Enclose the scratchpad between `<SCRATCHPAD>` and `</SCRATCHPAD>`. The scratchpad is secret and not seen by other parties. Your final answer is public and must never contain scores. Enclose your final answer after the scratchpad between `<ANSWER>` and `</ANSWER>`. Make your final answer very short and brief in 2-3 sentences and containing only your main proposals. Use options' short notations instead of long descriptions. Enclose any deals you suggest between: `<DEAL>` `</DEAL>`.

After the final answer, building on your current move and analysis, briefly write down short notes for yourself of what exact options you can explore the next time you speak. Enclose the notes between `<PLAN>` and `</PLAN>`.

Figure 12: Cooperative prompts structures - 5th Variation: CoT - No Previous Deals Calculations nor Candidate Generation

The following is a chronological history of up to [WINDOW SIZE] interactions <HISTORY> [HISTORY] </HISTORY>. The following are your previous plans from last interactions. You should follow them while also adjusting them according to new observations. <PREV PLAN> [LAST PLAN] </PREV PLAN>. Now it is your turn to talk.

Please use a scratchpad to explain yourself and why you are agreeing with a deal or suggesting a new one. In your scratchpad,

1) Based on the history and your notes, make a proposal that balances between your scores and accommodating others and that is more likely to lead to an agreement.

You must follow these important negotiation guidelines in all your suggestions:

- Aim for a balanced agreement considering all parties' interests.
- Show flexibility and openness to accommodate others' preferences.
- Express your objectives clearly and actively listen to others.
- Empathize with other parties' concerns to foster rapport.
- Focus on common interests to create a win-win situation.
- It is very important for you that you all reach an agreement as long as your minimum score is met.

Enclose the scratchpad between <SCRATCHPAD> and </SCRATCHPAD>. The scratchpad is secret and not seen by other parties. Your final answer is public and must never contain scores. Enclose your final answer after the scratchpad between <ANSWER> and </ANSWER>. Make your final answer very short and brief in 2-3 sentences and containing only your main proposals. Use options' short notations instead of long descriptions. Enclose any deals you suggest between: <DEAL> </DEAL>.

After the final answer, building on your current move and analysis, briefly write down short notes for yourself of what exact options you can explore the next time you speak. Enclose the notes between <PLAN> and </PLAN>.

Figure 13: Cooperative prompts structures - 6th Variation: CoT - No Previous Deals Calculations, Other's Preferences or Candidate Generation

