# OpenReview forum: "Reproducibility Study of "Cooperation, Competition, and Maliciousness: LLM-Stakeholders Interactive Negotiation""
_TMLR — Accepted by TMLR_

### Review · Reviewer_RQqG · 2025-03-13

**Summary Of Contributions:**

The paper explores the topic of evaluating the capabilities of LLM agents to work out a deal which is acceptable to most agents. A number of large and small language models are compared on a specific negotiation task, and a number of criteria and relevant metrics are used to discuss their performance in terms of information leakage and fairness of the deals. The work is closely related to a recent paper ("Cooperation, Competition, and Maliciousness: LLM-Stakeholders Interactive Negotiation" by Abdelnabi et al). The results show that Larger models tend to yield better performances, and single-agent systems may be able to match the performance of multi-agent systems on this task.

**Audience:**

Yes

**Claims And Evidence:**

Yes

**Requested Changes:**

I generally appreciate the work and don't have any particular change to suggest.

**Strengths And Weaknesses:**

The strength of the paper lies in the fact that it considers a number of aspects to evaluate the quality of the negotiations and final deals, and also make attempts to assess the fairness, accessibility and environmental impact of the approach. Furthermore, it also studies the single-agent version of the problem and obtains results that may match multi-agent systems.

One weakness of the paper is that the aim and analyses tend to adhere too closely to the paper "Cooperation, Competition, and
Maliciousness: LLM-Stakeholders Interactive Negotiation".

---

> ### Author Response · Authors · 2025-03-26
> **Summary of Changes**
>
> Dear RQqG,
>
>
> We thank you for your positive evaluation and thoughtful comments regarding our manuscript. We appreciate your recognition of our efforts to comprehensively evaluate negotiation quality, fairness, accessibility, and environmental impact, as well as our exploration of single-agent versus multi-agent systems.
>
> We acknowledge your observation regarding the close alignment of our aims and analyses with the paper we were reproducing. This was our initial intention for our reproducibility study, but we have included additional analyses in our revised manuscript.
>
> Although you did not request specific changes, we have implemented revisions based on feedback from other reviewers, which we believe have further strengthened the manuscript. We hope these improvements, shown in red in the current PDF version and listed below, will enhance the clarity and impact of our work:
>
> * Add experiments to test GPT-4's performance on the benchmark.
> * Include ablations to determine the prompt's impact on model performance.
> * Show results of how an adversarial party with veto power impacts the negotiation outcome.
> * Improved phrasing of motivation to use GPT-4o Mini instead of GPT-4 for most of our analyses.
> * Corrected description of Table 3 to reflect the right comparison to the original paper.
> * Excluded GPT-4o Mini from the plot comparing performance vs. model size.
>
> ---
>
> Thank you again for your valuable feedback.
>
> We are happy to address any questions or concerns you may have with the new version, so please let us know if you have further comments.

---

### Review · Reviewer_vekK · 2025-03-14

**Summary Of Contributions:**

This submission presents a reproducibility study of "Cooperation, Competition, and Maliciousness: LLM-Stakeholders Interactive Negotiation" and extension of its benchmark designed for evaluating LLMs in interactive negotiation scenarios, particularly focusing on cooperation, competition, and malicious behaviors. It reveals that larger models generally perform better, while smaller models often struggle with adhering to formatting rules and producing coherent responses. The study demonstrates that a single-agent baseline without inter-agent communication can achieve performance comparable to multi-agent interactions, challenging the necessity of communication in successful negotiations. Additional analyses include a Pareto efficiency study, structural leakage of information due to formatting errors, and the introduction of a fairness metric (Gini coefficient) to evaluate outcome equality. Ultimately, the study suggests that the benchmark primarily assesses an agent's ability to infer acceptable deals rather than genuine negotiation and communication skills, highlighting the importance of precise benchmarking methodologies in evaluating multi-agent interactions.

**Audience:**

Yes

**Broader Impact Concerns:**

No issues were founded.

**Claims And Evidence:**

No

**Requested Changes:**

- Please fix the two weaknesses mentioned above.
	- All experiments involving GPT-4o Mini need to be rerun using GPT-4, ensuring version consistency. Although I believe this won't significantly change the conclusions, the logical framework of the experiment design should be corrected.
- The description in Table 3 is incorrect. The Table 3 in this submission should correspond to Table 1 in the original paper, not Table 3.

**Strengths And Weaknesses:**

**Strengths**

- This is a work that challenges the reproducibility of other LLM articles. The current community needs this kind of spirit.
- The viewpoints are clearly expressed, making comparisons very convenient.
- The proposed new metrics make sense to me.

**Weakness 1**

In Section 4.1, the authors claim that:

> Furthermore, since the original paper focused on using GPT-4 (OpenAI, 2023), we chose GPT-4o Mini (OpenAI, 2024) as a more powerful and cost-effective alternative.

This is incorrect. The performance of GPT-4o Mini is far inferior to GPT-4. A model's power cannot be judged solely based on its release date. The key selling point of GPT-4o lies in the "o" standing for "omni," indicating its multimodal capabilities. But in pure text-based tasks, its performance falls short of GPT-4. The "Mini" label indicates that this model has undergone distillation, leading to a decrease in performance in exchange for a smaller parameter size and faster inference speed.

**Weakness 2**

The model version number is not explicitly stated, such as for GPT-4o Mini. Although OpenAI currently has only one version of this model, `gpt-4o-mini-2024-07-18`, new versions may be released in the future.

- One example is that GPT-4 has two versions: `gpt-4-0314` and `gpt-4-0613`.
- The timestamp should be mentioned in the main text.
- The code only calls `gpt-4o Mini` without specifying a particular historical version. This means it will always invoke the latest version of `gpt-4o Mini`. If a new version is released in the future, this may cause the author's code to behave differently from its current execution.

---

> ### Author Response · Authors · 2025-03-26
> **Summary of Changes Based on Review**
>
> Dear vekK,
>
> Thank you for your detailed review and constructive comments. We agree with the weaknesses you expressed in your review, so we have implemented the following changes in the manuscript:
>
> >This is incorrect. The performance of GPT-4o Mini is far inferior to GPT-4. A model's power cannot be judged solely based on its release date.
>
> >All experiments involving GPT-4o Mini need to be rerun using GPT-4, ensuring version consistency. Although I believe this won't significantly change the conclusions, the logical framework of the experiment design should be corrected.
>
> We agree that we misrepresented the performance of GPT-4o Mini compared to GPT-4. We found some benchmarks on our initial assessment ([1](https://llm-stats.com/models/compare/gpt-4-0613-vs-gpt-4o-mini-2024-07-18) and [2](https://lmarena.ai/) ) where GPT-4 is better than GPT-4o Mini and others where GPT-4o Mini has better performance, but we agree that it was necessary to use GPT-4 to confirm its metrics for this benchmark.
>
> We ran the base game experiments for the multi- and single-agent scenarios with GPT-4 and now present the results in Tables  2 and 5. We did not run all the other experiments with GPT-4 because both models perform similarly on the multi-agent benchmark (Table 2) and GPT-4o Mini performs better in the single-agent scenario (Table 5). Additionally, the cost of running an experiment with GPT-4 is around 100 times higher than with GPT-4o Mini. We believe this lower cost with comparable results is useful to increase the reproducibility of our results with a lower API budget.
>
> To reflect the nuance in the comparison between the two models, we changed the wording to motivate using GPT-4o Mini by arguing that it is 100x cheaper and performs similarly or better than GPT-4 in this benchmark.
>
> >The model version number is not explicitly stated, such as for GPT-4o Mini. Although OpenAI currently has only one version of this model, gpt-4o-mini-2024-07-18, new versions may be released in the future.
>
> We now provide the explicit versions for GPT-4o Mini and GPT-4 as a footnote in the paper and in the configuration files in the code. We agree that including the version is important for reproducibility purposes in the future.
>
> >The description in Table 3 is incorrect. The Table 3 in this submission should correspond to Table 1 in the original paper, not Table 3.
>
> We corrected the description for Table 3 to show it corresponds to Table 1 in the original paper.
>
> ---
>
> Thank you again for your detailed feedback and suggestions. We highlight all the changes in red for easier comparison. We agree these changes improve the rigor and clarity of our manuscript.
>
> We are happy to address any additional concerns you may have, so please let us know if you have further comments.

---

> > ### Comment · Reviewer_vekK · 2025-03-26
> >
> > I broadly agree with the authors' current position and the latest revisions to the text, based on the experimental results provided.
> >
> > **Issue 1**
> >
> > But I was surprised by the conclusion that `GPT-4o Mini` sometimes outperforms `GPT-4`. In Table 5, `GPT-4`'s single-agent performance appears somewhat anomalous. Do the authors have any hypotheses or insights into the possible reasons for this?
> >
> > **Issue 2**
> >
> > Aside from the single-agent setting, `GPT-4` generally performs better than `GPT-4o Mini`. The justification provided in the paper is that `GPT-4o Mini` is 100 times cheaper, which suggests a **trade-off**. If the reported results are reliable, I'm fine with not using `GPT-4` across the board. But from this cost-performance perspective, it seems natural to consider a middle-ground option—namely, `GPT-4o`. Did the authors consider including this model in their experiments?
> >
> > I'm not necessarily requesting that the authors conduct this additional experiment (`GPT-4o`), but it seems like a natural question to ask given the perspective presented. It may be worth further justification.

---

> > > ### Author Response · Authors · 2025-04-01
> > > **Response**
> > >
> > > Dear vekK,
> > > Thank you for reading our previous response. We appreciate the new round of comments and find our responses below:
> > >
> > > Issue 1:
> > >
> > > We looked into the single-agent’s performance for GPT-4, and the main conclusion is that GPT-4 is underestimating the needs of P2 or its veto power but accurately estimating the needs for other parties. P2 rejects the final deal with its veto power, and that is why the deals are unsuccessful. Compared to the multi-agent scenario, P2 cannot provide additional information to P1 during the interaction and help calibrate its needs to reach a successful deal in the single-agent setup. Without veto power of P2 most of the deals would be accepted as 5-way deals.
> > >
> > > The results below compare the outcome of the negotiation for GPT-4 and GPT-4o Mini for the single-agent setup. In the GPT-4 single-agent game there is a lot more of D1 in the final deals which lowers the score for P2 who holds the veto, but is more favourable for P1. This happens less for GPT-4o Mini, as can be seen from the results below, showing that GPT-4 underestimates the veto power rule of P2.
> > >
> > >
> > > | GPT-4           | single agent (6 rounds) | single agent (1 round) | GPT-4o Mini     | single agent (6 rounds) | single agent (1 round) |
> > > |----------------|--------------------------|--------------------------|------------------|--------------------------|--------------------------|
> > > |                | A2, B3, C2, D1, E3        | A2, B2, C3, D2, E5        |                  | A1, B2, C2, D2, E3        | A1, B2, C3, D2, E4        |
> > > |                | A2, B2, C3, D1, E2        | A1, B2, C3, D1, E4        |                  | A2, B2, C2, D1, E3        | A2, B2, C3, D2, E3        |
> > > |                | A2, B3, C2, D1, E2        | A2, B2, C3, D1, E5        |                  | A2, B2, C2, D2, E4        | A2, B2, C3, D2, E3        |
> > > |                | A2, B2, C3, D1, E4        | A2, B2, C3, D1, E4        |                  | A2, B2, C3, D2, E2        | A2, B2, C3, D2, E3        |
> > > |                | A2, B2, C3, D1, E4        | A2, B2, C3, D1, E4        |                  | A2, B2, C3, D2, E3        | A2, B2, C3, D2, E3        |
> > > |                | A3, B3, C2, D1, E1        | A2, B2, C3, D1, E4        |                  | A2, B2, C3, D2, E2        | A2, B2, C3, D2, E4        |
> > > |                | A2, B2, C3, D1, E4        | A1, B2, C3, D1, E4        |                  | A1, B2, C3, D2, E4        | A2, B2, C3, D2, E3        |
> > > |                | A2, B2, C2, D1, E2        | A2, B2, C2, D1, E3        |                  | A2, B2, C2, D1, E3        | A2, B2, C3, D2, E3        |
> > > |                | A2, B3, C3, D2, E4        | A2, B2, C3, D2, E3        |                  | A2, B2, C3, D2, E3        | A2, B2, C3, D2, E3        |
> > > |                | A3, B2, C3, D1, E3        | A2, B2, C3, D1, E4        |                  | A2, B2, C3, D2, E4        | A2, B2, C3, D2, E4        |
> > >
> > >
> > > Issue 2:
> > >
> > > We agree that GPT-4o, especially in its newest release, will probably provide a better tradeoff between performance and cost compared to GPT-4o Mini and GPT-4. We chose to only run with GPT-4 and GPT-4o Mini because the former was used in the original paper and the latter achieved a strong performance at a low cost. Since our focus was reproducing and evaluating the benchmark as a way to evaluate LLMs’ negotiation skills, we think we can leave running GPT-4o as a limitation and future work for this paper.

---

> > > > ### Author Response · Authors · 2025-04-10
> > > > **Updates on manuscript**
> > > >
> > > > Dear vekK,
> > > >
> > > > We updated the manuscript to include our response to issues 1 and 2 you mentioned in your previous comment. Specifically, we included the response to Issue 1 as Section A.3 in the Appendix and Issue 2 as a comment in the Limitations and Future Work section.
> > > >
> > > > Please let us know if you agree that the updates we made to the manuscript provide good evidence for the claims we make in the paper.

---

> > > > > ### Comment · Reviewer_vekK · 2025-04-10
> > > > >
> > > > > The overall recommendation was submitted on the same day I received the previous rebuttal, but I wasn’t aware that it was not visible to the authors. In my view, the issues I mentioned have now been resolved.

---

### Review · Reviewer_UjFx · 2025-03-15

**Summary Of Contributions:**

This paper is a reproducibility study and extension of “Cooperation, Competition, and Maliciousness: LLM-Stakeholders Interactive Negotiation.” It revisits the original multi-agent negotiation game where six LLM-based agents negotiate a multi-issue deal, reproducing the benchmark across multiple open-source models and a smaller closed-source alternative. The authors replicate the original findings (e.g., that larger models typically perform better and that prompting style can be critical) and propose additional analyses and metrics::

- A single-agent baseline demonstrating that one agent.
- Pareto-front analysis revealing that almost all “acceptable” deals lie on the Pareto front.
- A structural leakage metric to detect unintended revelation of private chain-of-thought.
- An inequality measure.

Ultimately, the authors argue that the current benchmark principally tests whether the main negotiator (p1) can produce any acceptable deal rather than actually measuring cooperative or communicative skills among agents.

**Audience:**

Yes

**Claims And Evidence:**

Yes

**Requested Changes:**

- The authors do an ablation study of chain-of-thought structures (Section 5.1 and Table 3 on pages 6–7). Building on that, further experiments could examine different types of system or role prompts to test if the results are stable to minor changes in prompt engineering. This would highlight generalizability.

- For a fair demonstration of adversarial or malicious behavior, consider adjusting the rules so that more than one agent can propose (or veto) a final deal. This change might produce stronger evidence of true sabotage or collusion and highlight meaningful differences between cooperating and adversarial runs.

- In Figure 1, gpt-4o-mini probably shouldn't be in it because its size is unknown.

**Strengths And Weaknesses:**

__Strengths__:
- Thorough replication and validation of original findings across models.
- Reveals that a single agent can perform comparably to multi-agent setups.
- Introduces structural leakage and inequality metrics for deeper analysis.
- Identifies and corrects bugs in the original implementation.
- Detailed discussion on smaller model failures and energy usage.

__Weaknesses__:
- While the authors present multiple outline variations in Table 3 (page 7) and see that prompt structure can critically affect results, the paper offers less discussion on how stable or robust these findings remain under further variations in system prompts or model hyperparameters.

- The paper reproduces targeted and untargeted adversarial scenarios (Tables 6–7 on pages 14–16), yet it acknowledges that adversarial agents may barely shift final deals if only p1 can propose. The interplay among adversarial models might be more convincingly explored in a setting where all agents’ votes or proposals matter.

---

> ### Author Response · Authors · 2025-03-26
> **Summary of Changes Based on Review**
>
> Dear UjFx,
>
> We sincerely thank you for your insightful comments and valuable suggestions to make our results more robust, as well as their recognition of our contributions.
>
> We have addressed your requested changes as follows:
>
> > The authors do an ablation study of chain-of-thought structures (Section 5.1 and Table 3 on pages 6–7). Building on that, further experiments could examine different types of system or role prompts to test if the results are stable to minor changes in prompt engineering. This would highlight generalizability.
>
> **Prompt Generalizability**: We conducted two additional experiments to examine the effect of different prompt structures that described the same game. First, we follow the original paper by using the base game rewritten by GPT-4 to have different parties but with the same incentives and score thresholds. Second, we use the same prompts from the base rewritten game but change the order of sections in generating the final prompt such as the: individual instructions, scoring rules, voting rules and incentive rules. We believe these variations show how the prompt structure affects the results. We present the results in Table 9.
>
> >For a fair demonstration of adversarial or malicious behavior, consider adjusting the rules so that more than one agent can propose (or veto) a final deal. This change might produce stronger evidence of true sabotage or collusion and highlight meaningful differences between cooperating and adversarial runs.
>
> **Adversarial Scenario Adjustment**: We add the results of setting P2 as the adversarial party in the negotiation. P2 has veto power on the final deal, so this setup provides stronger evidence of how an adversarial party may impact the outcome with a true power to sabotage the deal even if P1 makes the final proposal. We present the results in Table 6. We did not modify the game to allow more agents to veto, vote or propose a final deal because it significantly changes the original game dynamics—e.g., it changes the number of acceptable deals on the Pareto front. We agree with the reviewer that this is an interesting analysis and could be an extension of the negotiation benchmark, but we think it is out of scope for our analysis of the original paper.
>
> >In Figure 1, gpt-4o-mini probably shouldn't be in it because its size is unknown.
>
> **Figure 1 Correction**: We removed GPT-4o Mini and did not include GPT-4 in Figure 1, as their exact parameter size is unknown.
>
> ---
>
> All these changes are highlighted in red in the manuscript to facilitate your revisions.
>
> We appreciate your suggestions, which we believe have strengthened our manuscript.
>
> We are happy to address any additional concerns you may have, so please let us know if you have further comments.

---

### Author Response · Authors · 2025-04-10
**Summary of Updates**

Dear Reviewers and AE,

We thank you for your constructive comments and feedback about the paper we submitted. We wanted to end the review period by summarizing all the changes below:
- **Prompt Ablation Study**: We ran two experiments to examine the effect of different prompt structures on negotiation outcomes. The results are on page 7 and Table 10 in the Appendix.
- **Adversarial Scenarios**: We included the results of running P2 (the party with veto power) as an adversarial party to test how an adversarial party with veto power impacts the negotiation outcome. The results are in Table 6 and discussed on page 9.
- **Benchmark GPT-4**: We now include GPT-4 in Tables 2 and 5 to include the best-performing model from the original paper. We also include an additional section in the Appendix (A.3) to explain why GPT-4 performs worse than GPT-4o Mini in the single-agent scenario.
- **Model choice motivation**: We changed the phrasing explaining why we chose GPT-4o Mini instead of GPT-4 for most of the experiments to reflect the tradeoff between cost and performance. We also include GPT-4o in the Limitations and Future Work section as one model that could be relevant to test, given the tradeoff between performance and cost.
- **Model versions**: We make model versions explicit for GPT-4o Mini and GPT-4.
- **Table 3 correction**: We corrected the description for Table 3 to show it corresponds to Table 1 in the original paper.
- **Figure 1 correction**: Removed GPT-4o Mini and did not include GPT-4 in Figure 1, as their exact parameter count is unknown.

All changes are highlighted in red on the manuscript's latest version.

We believe your feedback led to changes that strengthen this work and make it more relevant to the TMLR audience.

---

### Decision · Action_Editor_W1r3 · 2025-04-07

**Recommendation:** Accept as is

**Comment:**

This paper is a reproducibility study of LLM negotiation systems. All three reviewers lean toward accepting it, noting that the research thoroughly reproduces the original work, corrects errors, and introduces new evaluation metrics. Overall, the reviews are positive, considering the work valuable to the LLM research community.

**Audience:**

yes

**Claims And Evidence:**

yes